


# Beachgoers' ability to identify rip currents at a beach in situ

Sebastian J. Pitman[1], Katie Thompson[1], Deirdre E. Hart[1], Kevin Moran[2], Shari L. Gallop[3,4], Robert W. Brander[5], and Adam Wooler[6]

[1]School of Earth and Environment, University of Canterbury, New Zealand
[2]University of Auckland, Auckland, New Zealand
[3]University of Waikato, Tauranga, New Zealand
[4]Environmental Research Institute, University of Waikato, Hamilton, New Zealand
[5]School of Biological, Earth and Environmental Sciences, UNSW Sydney, Australia
[6]Surf Life Saving New Zealand, Wellington, New Zealand

**Correspondence:** Sebastian Pitman (sebastian.pitman@canterbury.ac.nz)

**Abstract.** Rip currents ("rips") are the leading cause of drowning on surf beaches worldwide. A major contributing factor is that many beachgoers are unable to identify rip currents. Previous research has attempted to quantify beachgoers' rip spotting ability using photographs of rip currents, without identifying whether this usefully translates into an ability to spot a rip current in situ at the beach. This study is the first to compare beachgoers ability to spot rip currents in photographs and in situ at a beach

in New Zealand (Muriwai Beach) where a channel rip current was present. Only 22% of respondents were able to identify the in situ rip current. The highest rates of success were for males (33%), New Zealand residents (25%), and local beach users (29%). Of all respondents who were successful at identifying the rip current in situ, 62% were active surfers/bodyboarders and 28% were active beach swimmers. Of the respondents who were able to identify a rip current in two photographs, only 34% were unable to translate this into a successful in situ rip identification, which suggests that the ability to identify rip currents by

beachgoers is worse than reported by previous studies involving photographs. This study highlights the difficulty of successfully identifying a rip current in reality and that photographs are not necessarily a useful means of teaching individuals to spot rip currents. It advocates for the use of more immersive and realistic education strategies, such as the use of virtual reality headsets showing moving imagery (videos) of rip currents in order to improve rip spotting ability.

## 1 Introduction

Rip currents (colloquially known as "rips") are fast, narrow, seaward-directed flows of water that commonly exist on sandy beaches. Many different types of rip currents exist (Castelle et al., 2016), but one of the most common and best understood types are channel rips. Channel rips occupy morphologic depressions (channels) between adjacent sand bars and are generated by alongshore variability in wave breaking (Castelle et al., 2016), which provides a distinct visual signature (Figure 1). Rips originate near the shoreline and generally flow offshore, with typical mean flow velocities of 0.5 – 0.8 ms-1 and instantaneous

velocities occasionally reaching 2 ms-1 (MacMahan et al., 2006). It is therefore not surprising that rip currents are regarded as the primary surf zone hazard for bathers and swimmers on beaches where they exist (Brander and Scott, 2016). The lack of national reporting structures for drownings mean that the true extent of global rip related drowning is unknown (Brander and





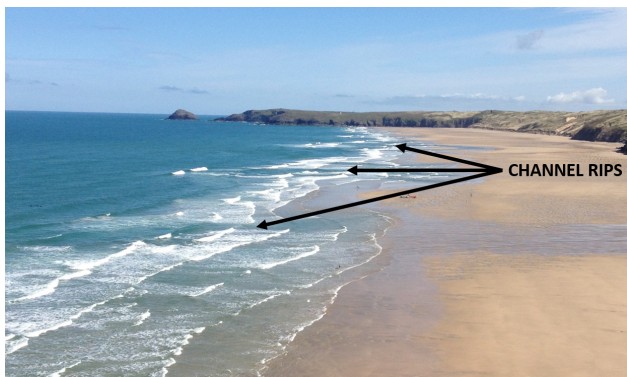

**Figure 1.** Channel rips typically present as darker corridors of water between the whiter patches of breaking waves, as shown here at Perranporth Beach, UK. The rips occur in deeper channels which reduces or prevents wave breaking, hence the darker visual signature. The rip channels often look like the calmest or safest place to swim to inexperienced observers. [Image credit: SP].

MacMahan, 2011), but some notable studies estimate in excess of 100 drownings per year in the United States (Brewster et al., 2019), and around 21 per year in Australia (SLSA, 2019b).

The degree of physical hazard that rip currents represent is controlled largely by the temporal and spatial variability in their occurrence, flow velocity and flow circulation patterns (Scott et al., 2014; Pitman et al., 2016; Gallop et al., 2018). However, the risk or likelihood of a rip current-related drowning or rescue occurring also depends on multiple social factors, such as the presence/absence of lifeguards, choice of swim location, the number of beachgoers and water users, water competency, beachgoer behaviour, and their knowledge of rip currents (Gilchrist and Branche, 2016; Ménard et al., 2018).

There are several initiatives and interventions employed around the world to mitigate the social aspects of rip current drowning risk. It is well established that lifeguards are the most effective method for drowning prevention on popular surf beaches (Gilchrist and Branche, 2016). However, the coverage of lifeguarding services varies spatially and temporally. Logistical and cost constraints, as well as seasonality of demand, mean it is not feasible for lifeguards to be present on all beaches and at all times. For example, in New Zealand there are 74 Surf Life Saving Clubs spread around 15,000 km of coastline, and their

patrol season typically only runs from October to April surrounding the Austral summer. Furthermore, the lifeguard beach flag systems used globally are inconsistent, varying from the traffic light system approach of the United States to the 'swim between the red and yellow flag' system adopted by Australia, New Zealand, the UK and some other countries. Beach safety signage is another commonly adopted mitigation method used to educate people about the rip current hazard, but the type and messaging involved varies globally. There is also evidence that signage at entry points to beaches goes unnoticed by a large proportion

of beach users (Matthews et al., 2014), and that signage is often ineffective in communicating key messages to beachgoers (Brannstrom et al., 2015).

    More recently, a number of dedicated national education campaigns have focussed specifically on rip current hazard interventions. For example, the United States 'Break the Grip of the Rip!' campaign (www.ripcurrents.noaa.gov), which began in 2005 focussed on a wide range of factors, such as how rip currents operate, why they are dangerous, how to spot them, and





how to escape them (Carey and Rogers, 2005), and this campaign has been shown to be somewhat effective (Houser et al., 2017). In Australia, several campaigns since 2009 have focussed on how to escape rip currents, how to spot them and more recently the 'Think Line' campaign has been adopted (www.beachsafe.org.au/surf-safety/ripcurrents), where beachgoers are urged to stop, observe, and think about what the hazards are when they got to a beach. Similarly, the UK has the 'Respect the Water' campaign (www.respectthewater.com/), which focuses more broadly on raising awareness of the wider dangers

associated with coastal or ocean recreation. New Zealand has also recently released the 3Rs ('Relax, Raise, Ride') rip current campaign (www.findabeach.co.nz/besafe/hazards/rips/), aimed at reminding people to remain calm and conserve energy in order to remain afloat, whilst signalling for help, going with the current, and weighing up your options to get back to shore.

One of the key themes in many existing rip current safety campaigns has been attempting to teach people to identify a rip current. However, several studies have shown that even when people are aware of what rip currents are and why they

are dangerous, they are largely unable to visually identify a rip current in a photo. Caldwell et al. (2013) reports successful identification of channel rips to be lower than 20%, with success rates of up to 40% reported by Willcox-Pidgeon et al. (2017), and a 48% success rate in selecting a safe place to swim reported by Sherker et al. (2010). A similar study by Surf Life Saving Australia (SLSA, 2019a) surveyed ocean swimmers who self-assessed as highly competent, and reported that only 27% were able to accurately identify rip currents in multiple photographs. Evidence suggests a similar trend in other rip current types,

with a study by Brannstrom et al. (2014) reporting 31% success when beachgoers were asked to identify a boundary-controlled rip current running along the edge of a groyne.

Rip spotting is complicated by the fact that different rip current types or forcing conditions can create different visual signatures. For example, channel rips that form in deeper channels between sand bars (Castelle et al., 2016) generally present optically as an area of darker and calmer water due to a relative lack of wave breaking when compared to the shallow sandbars

either side (Figure 1). Conversely, flash rips (Castelle et al., 2016) are not channelized, but instead generated by transient surf zone eddies resulting from vortical motions associated with short-crested breaking waves, are typically characterised by sediment-laden plumes of water extending offshore and a turbulent water surface. The wave-current interaction between incoming waves and the offshore rip current flow can also present visually as a rippled and bumpy water surface (Ménard et al., 2018).

The calm, smooth visual signature associated with channel rips is particularly important in terms of educating people about how to identify the hazard as inexperienced or uninformed beach users will often pick this calmer patch of water as the safest place to swim (Gallop et al., 2016) to avoid the breaking waves either side of the rip, which are perceived as being more dangerous (Caldwell et al., 2013). Rip current visual signatures also vary as morphology, tidal stage, or wave energy changes (Pitman et al., 2016), meaning an observer might have to rely on different visual signatures for the same rip channel at different

times during the day.

The ability to spot, and therefore avoid, a rip current is a critical skill for a beachgoer when making decisions about where and when to enter the water, particularly in the absence of lifeguards (Ménard et al., 2018). As demonstrated above, previous studies have shown that existing rip current identification abilities of beachgoers is poor. However, in these studies, rip current identification by beachgoers has either been based on participants self-reporting on how confident they would feel at being





asked to identify a rip current (Morgan et al., 2009a), or by asking people to directly identify rip currents in photographs taken from various perspectives (Moran, 2008; Caldwell et al., 2013; Brannstrom et al., 2015; Clifford et al., 2018), or to identify the safest place to swim in a photograph of a beach (Sherker et al., 2010; Gallop et al., 2016; Houser et al., 2017; Willcox-Pidgeon et al., 2017; Clifford et al., 2018; Fallon et al., 2018). However, Ménard et al. (2018) noted that a fundamental problem is the lack of research investigating whether an individual's ability to identify a rip, or a safe swimming area in a

photograph, translates to an equal ability to spot a rip current in situ at the beach. Therefore, the aims of this study are to use a survey instrument to investigate: (1) how an individual's demographic (e.g. gender, age, ethnicity) and beach competence (e.g. swimming ability and degree to which they are familiar with the surf zone) relates to their ability to identify a rip current in situ at a high energy beach; and (2) whether the ability to spot a rip in a photograph translates to an equal ability to locate a rip in situ at a beach.

## 2    Study site and methods

### 2.1    Study site

The study site was Muriwai Beach (Figure 2), a high-energy, dissipative, mesotidal beach on the exposed West Coast of New Zealand's North Island. An analysis of wave hindcast data shows average significant wave height at Muriwai to be 2.1 m, with mean wave period of 10 seconds. Waves during the summer months are typically calmer than in winter, but are interspersed

with very high energy events associated with ex-tropical cyclone activity in the Tasman Sea. The surf zone typically exists in a double barred state, with a dissipative outer bar and intermediate inner bar (Brander and Short, 2000). This site was selected for this study as it is a high-risk site for rip current rescues according to Surf Life Saving New Zealand (SLSNZ), with 530 such events recorded in the period 2007 – 2018, representing 80% of beach lifeguard rescues at this site. Muriwai Beach is less than an hour's drive from New Zealand's largest city, Auckland (Figure 2a), making it a popular destination for both domestic

visitors and foreign tourists. Analysis of lifeguard headcount data show that visitor numbers at Muriwai Beach typically exceed 80,000 per month in the peak summer period. During the study period, lifeguards patrolled a safe bathing area indicated by a pair of red and yellow flags, located between a headland at the southern end of the beach, and a prominent channel rip to the north of the bathing area, in front of the lifeguard tower (Figure 2b). This rip channel was static throughout the 7 day study period.

### 2.2    Beach survey design

The research relied on a survey instrument completed by various beach users who were either on the beach directly onshore from the channel rip evident in Figure 2b, or in the flagged bathing area adjacent to it. The survey was approved by the University of Canterbury human ethics committee (2018/97/LR), and conducted over a 7-day period in January 2019 (Austral Summer) incorporating both weekdays and the weekend. Potential participants were approached by the investigator if they

were settled on the beach (i.e., not if they had just got there, or were just leaving), and if they appeared to have no supervisory





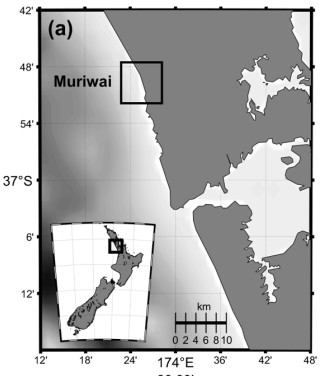 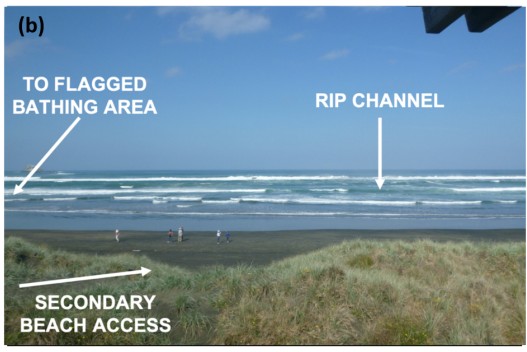

**Figure 2.** (a) Study site map and (b) photo of main channel rip next to bathing area, used to test rip identification ability. Photo is taken by the lifeguard tower, next to a secondary access track linking the car park to the beach.

duties such as watching young children in the water. The survey took less than 10 minutes to complete, with participants self-completing their answers on a paper form (see supplementary material). Additionally, one follow up question was presented verbally by the investigator, where the participants were asked whether they could see any rip currents at this site, and if so, to point towards and describe the location. The survey consisted of 34 questions, grouped broadly into basic demographic

questions (gender, age, ethnicity, and a question to identify whether the individuals were locals, wider New Zealand residents, or international visitors), as well as questions to ascertain how the participants spend their time at the beach, their swimming ability, and their understanding of rip current hazards. Answers fields in the survey made use of a combination of: (1) categorical tick boxes for questions about demographics, swimming ability or type of activity undertaken; (2) five-point Likert scales for questions addressing self-reported confidence in dealing with hazards; and (3) open text fields to understand the depth and

breadth of knowledge of participants with respect to different hazards. A selection of example questions is outlined in Table 1 and the full questionnaire is included as supplementary material.

This survey used Protection Motivation Theory (PMT; Rogers, 1975) to investigate rip current hazard perception, in line with previous water-based studies of competency (McCool et al., 2008, 2009; Moran et al., 2011, 2018; Moran and Willcox, 2013; Willcox-Pidgeon et al., 2018). The use of PMT allowed us to quantify and compare the self-reported perceptions of water

competence (including swimming and floating competency), versus the perceptions of risk in the surf zone. Water competence (as opposed to swimming ability) refers to a broad set of skills important in drowning prevention, with a full review of water competencies provided by Stallman et al. (2017). Participants were first asked if they could swim (yes/no), and if they answered yes, were then asked to rate their ability on a four-point scale from poor to very good. This was then quantified using questions asking how far they could swim in a pool, how confident they felt swimming that same distance at the beach, and when they

last swam that distance. This was followed with basic competence questions identifying their perceived ability to float and/or swim effectively on both their front and back, and their ability to tread water for two minutes.



**Table 1.** Question groups and example questions used in the survey of beach users

| Group | Focus of questions | Example question |
|---|---|---|
| 1 | Demographics | Do you live in New Zealand? |
| | | How old are you? |
| 2 | Recreational use of the beach | Is this your local beach? |
| | | How often do you visit beaches in the Summer? |
| | | When you visit the beach which activities do you undertake? |
| 3 | Water competence | Can you swim? |
| | | How far could you swim in a pool without stopping? |
| | | When did you last swim this distance? |
| | | How do you feel about swimming that same distance in the sea? |
| 4 | Beach hazards | What do the red/yellow flags mean on a New Zealand beach? |
| | | When would you swim outside the patrolled area? |
| | | Could you spot a rip current? |
| | | If asked to identify a rip current, what would you look for? |
| 5 | Rip identification | Put a mark anywhere you think you can see a rip current in the images below |
| 6 | Education | Have you had the opportunity to learn about rip currents before? |
| | | If yes, how did you learn about rip currents? |

Respondents were then asked open questions about how they select safe areas to swim at the beach, when and why they might choose to swim outside of the flagged bathing area, what hazards may occur at the beach, and which hazards they have personally experienced (Table 1). They were then specifically asked about rip currents, such as what makes a rip current

dangerous, how confident they feel about escaping a rip current, whether they have any experience of being caught in a rip, and the actions they would take to escape. Finally, we quantified their ability to spot rip currents using two photographs (shown in Figure 3), a method modified from that of Caldwell et al. (2013). Participants were then asked to identify a rip current on the beach in front of them. The presence of an active rip current was decided by consultation between a senior lifeguard, and the investigator who is experienced in surf zone dynamics, and provided both agreed that a channel rip was present and visible at

the time of the survey, the question was asked of the participant. Examples of times when the question was not asked include mid- to high-tide, where there was insufficient wave breaking on the inner bar to establish rip current circulation, or if the participant refused. If the participant was unsure how to identify the rip, or incorrectly identified an area where no rip was present, the response was marked as incorrect.





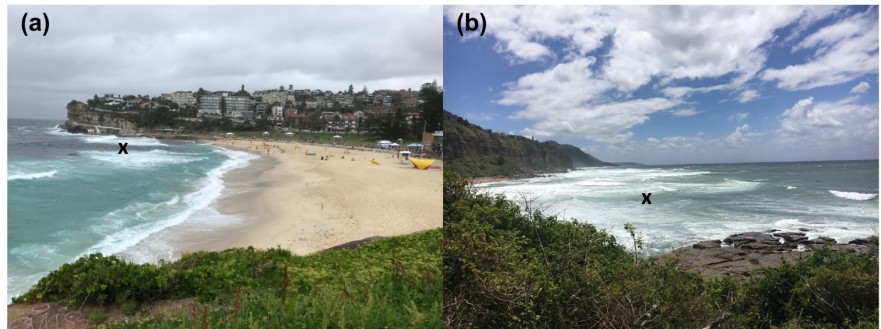

**Figure 3.** Images from (a) Bronte Beach and (b) Coalcliff Beach in New South Wales, Australia were used for checking participants' ability to identify rip currents. These were chosen as they contained rip channels that presented visually as darker areas between breaking waves, which is similar to the rip current present at the study site during the surveys. Participants were asked to mark with an 'x' any area they thought there was a rip; an example identification is shown above. Photo credits: (a) Walkingmaps.com.au; (b) Rob Brander.

## 3   Results

There were 132 surveys conducted whilst the rip current was active and displaying a visual signature that could be observed from the beach. Descriptive statistics for the sample are provided in Table 2. The sample population is gender-imbalanced, with 65% of respondents being female. Fifty four percent of the sample were New Zealand European, 17% were European, and 8% were Māori or Pacific Islander. Three percent of respondents were Chinese, 3% were Indian, and the remaining 15% were a combination of 'Other' nationalities. Eighty one percent of respondents were living in New Zealand, with 18% representing

holidaymakers or short-term visitors, and 31% of respondents classified the survey site as their local beach. Seventy four percent of respondents indicated that they go to the beach at least once a week, with 13% indicating they usually visit every day. Respondent age was recorded in discrete groups, with the modal group being 20 – 29 years (30%). Twenty two percent were aged between 10 and 19 years, 18% were 30 – 39 years, and 11% were 40 – 49 years. Seventy five percent of the sample indicated that they could swim in excess of 50 m in a pool.

## 3.1   Demographic trends in rip spotting ability


Seventy eight percent of respondents ($n$ = 103) were unable to spot the in-situ rip current. Table 3 shows how reported demographics influence the ability to spot the in situ rip. A significantly higher proportion of males (33%, $n$ = 15) than females (16%, $n$ = 14) were able to spot the rip ($\chi2$ = 4.66, p = 0.03). New Zealand Europeans had the highest successful rip identification rate (31%, $n$ = 22). Chinese and Indian respondents both had a 25% success rate, but the sample size for each ethnicity was

only 4. Māori and Pacific Islanders had a 20% ($n$ = 2) success rate. Success rate was higher in New Zealand residents (25%, $n$ = 27) than visitors (8%, $n$ = 2). No specific trends were evident with regard to respondent's age and ability to identify the rip current. Most drowning literature identifies those under 30 as most likely to be involved in a rip current rescue. When grouped





**Table 2.** Sample characteristics in terms of demographics, frequency of beach visits, and self-rated pool swimming ability (*n* = 132 ).

| Question | Response | n | % |
|---|---|---|---|
| Gender | Female | 86 | 65 |
| | Male | 46 | 35 |
| Ethnicity | New Zealand European | 71 | 54 |
| | European | 25 | 19 |
| | Other | 18 | 14 |
| | Maori or Pacific Islander | 10 | 8 |
| | Chinese | 4 | 3 |
| | Indian | 4 | 3 |
| Do you live in New Zealand? | No | 24 | 18 |
| | Yes | 107 | 81 |
| Age | 0 - 10 yrs | 2 | 2 |
| | 10 - 19 yrs | 29 | 22 |
| | 20 - 29 yrs | 40 | 30 |
| | 30 - 39 yrs | 24 | 18 |
| | 40 - 49 yrs | 15 | 11 |
| | 50 - 59 yrs | 10 | 8 |
| | 60 - 69 yrs | 5 | 4 |
| | >70 yrs | 2 | 2 |
| Frequency of beach visit | Daily | 17 | 13 |
| | 2 - 3 times per week | 42 | 32 |
| | Once per week | 39 | 30 |
| | Once per month | 13 | 10 |
| | Infrequently | 20 | 15 |
| Pool swimming ability | <25 m | 9 | 7 |
| | 25 - 50 | 22 | 17 |
| | 51 - 100 | 30 | 23 |
| | 101 - 200 | 22 | 17 |
| | 200 + | 47 | 36 |

here, those 29 and under had a success rate of 21% (*n* = 15), and those 30 and over had a success rate of 25% (*n* = 14), again showing no significant difference.





**Table 3.** Respondents ability to spot an in situ rip based on their demographic (*n* = 132).

| Question | Response | Unable to spot rip | | Able to spot rip | |
|---|---|---|---|---|---|
| | | n | % | n | % |
| Gender | Female | 72 | 83.7 | 14 | 16.3 |
| | Male | 31 | 67.4 | 15 | 32.6 |
| Ethnicity | New Zealand European | 49 | 69.0 | 22 | 31.0 |
| | European | 24 | 96.0 | 1 | 4.0 |
| | Other | 16 | 88.9 | 2 | 11.1 |
| | Maori or Pacific Islander | 8 | 80.0 | 2 | 20.0 |
| | Chinese | 3 | 75.0 | 1 | 25.0 |
| | Indian | 3 | 75.0 | 1 | 25.0 |
| Do you live in New Zealand? | No | 22 | 91.7 | 2 | 8.3 |
| | Yes | 80 | 74.8 | 27 | 25.2 |
| Age | 0 - 10 yrs | 2 | 100.0 | 0 | 0.0 |
| | 10 - 19 yrs | 21 | 72.4 | 8 | 27.6 |
| | 20 - 29 yrs | 33 | 82.5 | 7 | 17.5 |
| | 30 - 39 yrs | 20 | 83.3 | 4 | 16.7 |
| | 40 - 49 yrs | 10 | 66.7 | 5 | 33.3 |
| | 50 - 59 yrs | 8 | 80.0 | 2 | 20.0 |
| | 60 - 69 yrs | 3 | 60.0 | 2 | 40.0 |
| | >70 yrs | 1 | 50.0 | 1 | 50.0 |

## 3.2 Self-reported swimming competence

Respondents were asked to self-rate their swimming competency through a series of questions (Table 4), with each response assigned a numerical value between 1 and 5, representing increasing competence. When asked to rate on a qualitative spectrum ('Poor', 'Fair', 'Good', 'Very Good'), most respondents replied positively and there was little difference between genders with 68% of females and 69% of males indicating their swimming competence to be either good or very good. PMT requires this assertion to be qualified, and so they were subsequently asked to translate this into their pool swimming competence. In response, 47% of females reported being able to swim in excess of 100 m, compared to 61% of males. When subsequently asked how confident they would feel about swimming that same distance in the sea, only 32% of females responded positively (confident, or very confident), compared to 54% of males. No statistical difference was evident in self-reported estimates of swimming competence and confidence by gender. There was a significant difference between female ($x^-$ = 3.06; $\sigma$ = 1.11) and male ($x^-$ = 3.67; $\sigma$ = 0.86) responses to the question about swimming the same distance at sea (p = 0.001), with males reporting a value 0.61 higher than females, 95% CI [0.26 – 0.96].




**Table 4.** Self-reported swimming competency, broken down by the percentage of male and female respondents. The value in brackets was an assigned numerical value to aid statistical analysis of responses.

| Question | Response (Value) | % Female respondents | % Male respondents |
|---|---|---|---|
| Rate your swimming competency | Poor (1) | 2 | 4 |
| | Fair (2) | 28 | 24 |
| | Good (3) | 47 | 41 |
| | Very good (4) | 21 | 28 |
| How far could you swim in a pool? | <25 m (1) | 8 | 4 |
| | 25 – 50 (2) | 19 | 13 |
| | 51 – 100 (3) | 23 | 22 |
| | 101 – 200 (4) | 17 | 15 |
| | 200 + (5) | 30 | 46 |
| How do you feel about swimming that same distance in the sea? | Very anxious (1) | 8 | - |
| | Anxious (2) | 21 | 7 |
| | Unsure (3) | 34 | 37 |
| | Confident (4) | 22 | 37 |
| | Very Confident (5) | 10 | 17 |

Significant differences were evident when ability to identify an in situ rip were analysed against self-reported estimates of swimming competence (Figure 4a) and maximum pool swim distance (Figure 4b). A two sample t-test showed that the ability to identify an in situ rip current was higher among those that reported higher competence. The mean values were 0.34 (95% CI 0.02 – 0.67, p = 0.04) and 0.76 (95% CI 0.23 – 1.28, p = 0.006) higher in the self-estimated swimming competence and max pool swim distance categories respectively, for participants that successfully identified the rip current.

### 3.3 Familiarity, behaviour and experience

The questionnaire was able to ascertain how familiar people were with the study site, and what degree of interaction they had with the water. Of the 29 people able to spot the in situ rip current, 27 (93%) lived in New Zealand. Twenty two (92%) of the 24 people who lived overseas were unable to spot the rip. Twelve (41%) of the 29 people able to spot the rip classed Muriwai as their local beach, which equated to a 29% success rate among locals (n = 41).

Respondents were asked to state all activities they participate in at the beach, and these were ranked according to increasing interaction with the surf zone, from those that remain on the beach, those that enter the water, but remain in water shallow enough that they can stand, those that swim beyond their depth, to those that surf or body board (Figure 5a). Of the 29 respondents able to spot the rip current, the largest proportion (62%, n = 18) were in the surfing and body boarding group, followed by swimmers (28%, n = 8), and those that remain within their depth in the water (10%, n = 3). Of the 5 respondents

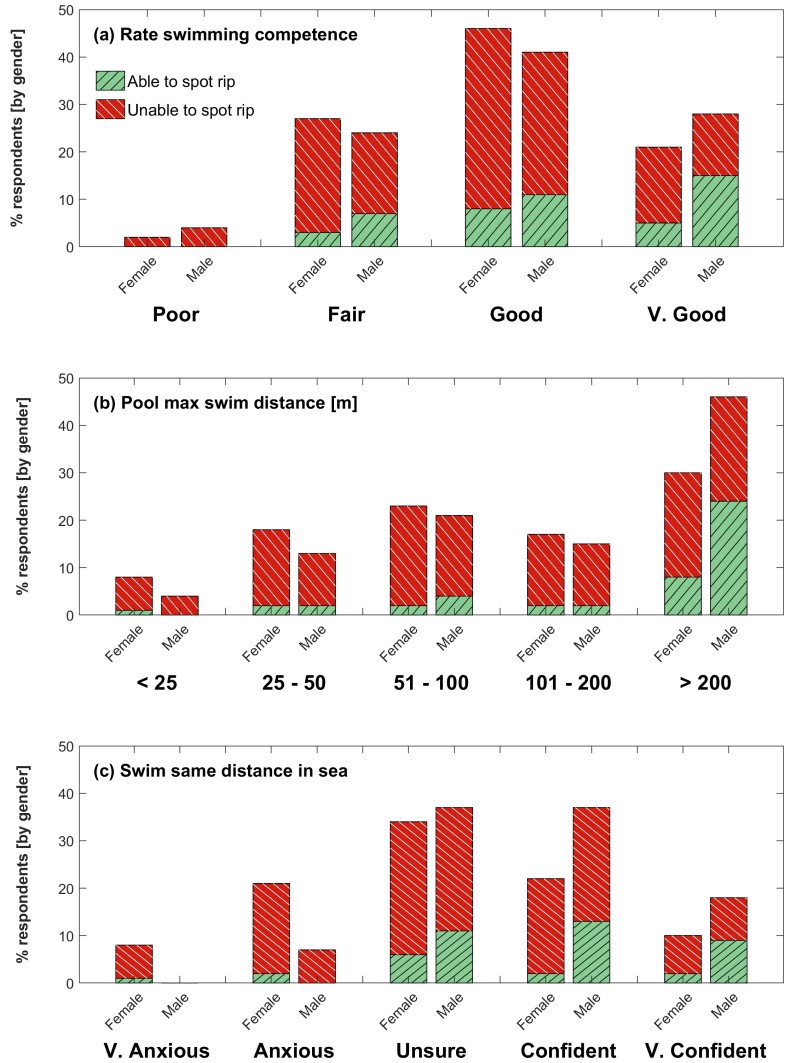

**Figure 4.** Respondent's ability to spot the in situ rip current as a function of: (a) their self-reported swimming competence; (b) the maximum distance they reported being able to swim in a pool non-stop; and (c) their feeling at being asked to swim the same distance in the sea. Responses are broken down by gender, with green bars indicative of the percentage of respondents able to spot the in situ rip, and red bars indicative of those that cannot.

who reported never entering the water at the beach, none were able to spot the rip current, although 4 out of the 5 did rate themselves as fair or good swimmers, so their decision to not enter the water did not appear to be a reflection of their perceived low swimming competence. In the swimming group, 85% ($n = 46$) of respondents were unable to identify the rip, as were 67% ($n = 37$) of the surfers and body boarders.

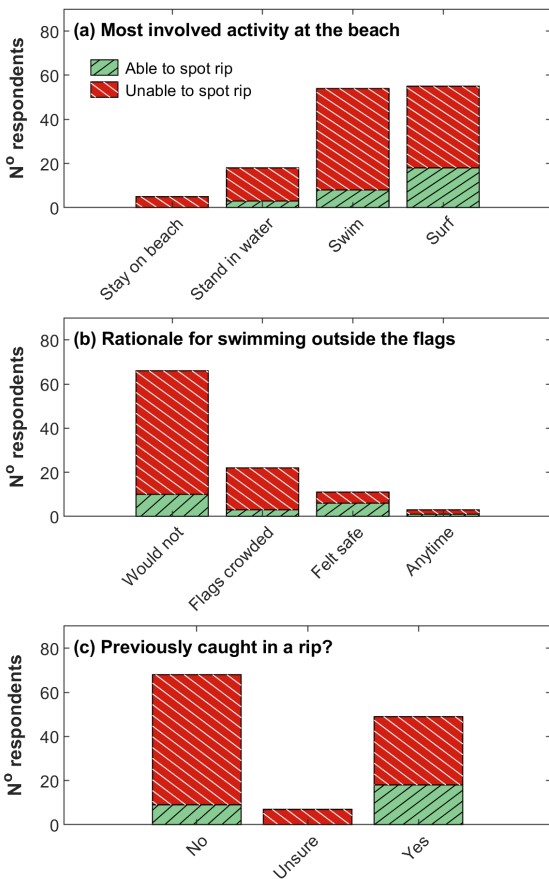

**Figure 5.** (a) Respondent's activity at the beach; (b) rationale for swimming outside of the flagged bathing are; and (c) indication of those who have experience of being caught in a rip. Green bars are indicative of the number of respondents able to spot the in situ rip, and red bars indicative of those that cannot.

When questioned on whether they would swim outside of the flagged bathing area, 35% (*n* = 36) of respondents indicated that they would at times swim outside of the patrolled area at the beach. Of these respondents, 72% (*n* = 26) were also unable to spot the rip current (Figure 5b). The reasons given for swimming outside the patrolled area were that the flags were too crowded (61%, *n* = 22), they felt able to choose a safe place to swim (31%, *n* = 11), or they didn't have a specific reason and 200 would do so at any time (8%, *n* = 3). The most competent subgroup in terms of rip spotting ability were those that felt they could identify a safe place to swim, in which 6 of the 11 (55%) were able to identify the rip. Across the entire dataset, the majority of respondents (55%, *n* = 68) indicated that they had not been caught in a rip before, and of these, 87% (*n* = 59) were unable to identify the rip (Figure 5c). Of the 49 respondents who had experienced being caught in a rip, only 37% (*n* = 18) were able to identify the in situ rip.

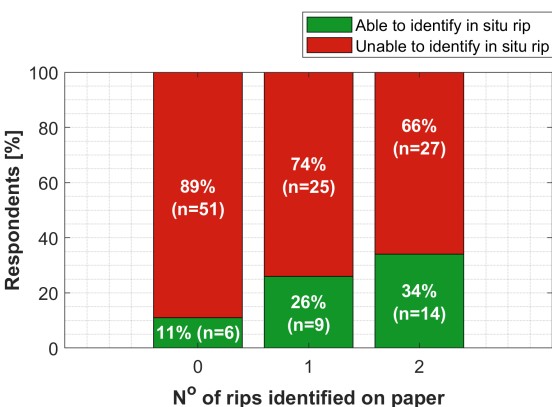

**Figure 6.** The percentage of respondents able to identify the in situ rip (green bars) versus those unable to identify the in situ rip (red), presented as a function of their ability to identify rips on paper.

## 3.4 Photo versus in situ identification

An implicit assumption of previous studies relating to rip identification is that the ability to spot a rip current in a photo translates to an ability to identify rip currents in situ. Figure 6 identifies the success rate of in situ rip identification as a function of participant's ability to identify rips on paper. Of the participants that were unable to identify a rip current in either of the images presented in Figure 3, 89% ($n$ = 51) were unable to identify the in situ rip. Of those able to identify the rip in only one of the images in Figure 3, 74% ($n$ = 25) were unable to identify the rip in situ, and of those able to identify rips in both images in Figure 3, only 34% ($n$ = 14) were able to identify rip currents in situ (Figure 6). The significance of the relationship between in situ and photo-based ability was tested using a linear regression, where the ability to spot a rip in situ was coded as a binary outcome, and the number of rips identified on paper was used as a continuous predictor. The number of rips identified in photos was a statistically significant ($p$ = 0.004) coefficient in the prediction of a participant's ability to identify in situ rips. Despite this statistical significance, it is important to acknowledge that approximately two thirds of respondents who were able to identify both rip currents in photographs, were unable to do so in situ at the beach (Figure 6).

## 4 Discussion

This study, to our knowledge, represents the first attempt to quantitatively evaluate the ability of beachgoers to visually identify an actual rip current on a beach. Only 22% of surveyed beachgoers were able to successfully spot a channel rip at a high energy beach at Muriwai, New Zealand. Here we discuss the results of our study in the context of existing literature in regards to both beachgoer demographics and implications towards beach safety.




## 4.1 Demographic trends in rip spotting ability

Males are generally over-represented in rip-related rescue statistics (Woodward et al., 2013) and global drowning epidemiology more generally (Peden and McGee, 2003), and PMT shows them to be more likely to overestimate their ability and under-
estimate the risk they are in with regard to water-related recreation (Moran, 2011). In the current study, males generally self-reported higher competence, with men significantly more likely to report feeling confident at the prospect of swimming their maximum pool distance at sea. This compares with previous studies identifying higher self-reported confidence in males, which is likely to increase drowning risk exposure through reduced inhibitions around swimming in deep water or challenging conditions (Morgan et al., 2009a). Males were better able to identify the rip current, but this result was not controlled for other
factors such as experience or familiarity with the beach in question. It is possible that the correlation with gender is linked to the fact that males are more likely to use surf equipment and swim further from the shore (Morgan et al., 2009b), and therefore are more likely to have frequently interacted with rip currents. A previous New Zealand study reported 53% of males felt confident identifying rip currents compared to 39% of females, but it did not report on their actual success rate (Moran and Ferner, 2016). Other studies addressing the impact of gender on the ability to identify a rip have had contrasting outcomes. For
example, males at Miami Beach, USA, were more successful in identifying a safe spot to swim (Fallon et al., 2018), whereas females were more successful in an Australian study (Williamson et al., 2012). Higher self-reported competence across both genders was linked to increased ability to spot the rip current in our study.

Respondents who undertook beach activities where they physically interacted more with the surf zone, such as swimming and surfing, were more likely to be able to spot the rip current. Surfing as an activity is associated with the type of coastline where
many hazards may exist, such as larger waves and stronger rips. Therefore, surfers likely have a much higher understanding of rip currents (Attard et al., 2015), especially as surfing is often prohibited in the flagged bathing area which increases the likelihood of interacting with a rip current. Moreover, many surfers actively use rip currents to get out beyond the breakers with minimal effort, and are therefore more adept at identifying them. Prior experience of being involuntarily caught in a rip (as opposed to choosing to use one) was also a factor in whether a person could identify the rips in this study. This was consistent
with an Australian study of rip current survivors showing that 84% of those people that had previously been caught in a rip current were now able to identify rips in photos (Drozdzewski et al., 2012). In our study, only 37% of rip current survivors were able to identify the in situ rip, which highlights the additional complexity and skill required when considering an active and fluid surf zone, rather than a static photo. This was evidenced in the study by Sherker et al. (2010), where 93% of respondents indicated they could identify a rip when in reality less than two thirds of the respondents could actually identify the channel rip
in a photo.

## 4.2 Implications for beach safety

As mentioned previously, only 22% of surveyed beachgoers in this study were able to identify a real rip current. Furthermore, 66% ($n = 27$) of the surveyed beachgoers that were able to successfully identify rip currents in two photos (Figure 3) were unable to identify the channel rip present at Muriwai Beach. These findings have significant implications for beach safety





practitioners on several levels. First, the ability to correctly identify rip currents on paper (i.e. still images) may result in overconfidence of actual rip identification ability, and therefore may lead to more risk taking behaviour, such as swimming away from lifeguards or in unpatrolled locations. Anecdotal evidence from Surf Life Saving Australia (Daw, 2019) showed an increase proportion of rescues of people who were considered educated and informed about rip currents following recent education campaigns, who, armed with this information, were now over predicting their ability to spot, avoid and escape rip

currents, and under predicting the risk.

Second, it suggests that the ability of beachgoers to identify rip currents may be worse than previously shown in the literature. The main methodological approach in previous studies has been either to directly ask surveyed participant to identify a rip current in an image (e.g. Brannstrom et al., 2014; Clifford et al., 2018) or to ask them to identify the safest place to swim (e.g. Sherker et al., 2010; Gallop et al., 2016; Warton and Brander, 2017) in an image that contained a rip current. The ability of

beachgoers to specifically identify rip currents has generally resulted in successful identification rates less than 30%. Caldwell et al. (2013) reported that less than 20% of participants were able to identify the channel rip in a series of images of Pensacola Beach, Florida, under green, amber and red flag conditions, with many instead thinking the rip current was present in the heavy surf which was instead indicative of shallower depths and wave breaking. Brannstrom et al. (2014) showed multiple images of a groyne at Galveston Beach, Texas, under different wave conditions and asked participants to identify the most hazardous

conditions to swim in. Only 13% of respondents identified the image with the boundary rip current, with the majority instead also opting for images of wave breaking. Similar results have been obtained from studies asking participants to identify safe swimming spots. Sherker et al. (2010) showed that 52% of primarily Australian respondents chose a rip current as the safest place to swim in an image, while 40% did so in the UK based study of Gallop et al. (2016), and 73% chose a rip current as the safest place in at least one of the two images they were shown in a study at Miami Beach, Florida, by Fallon et al. (2018).

While our finding that only 22% of respondents could identify the in situ rip is therefore at the lower end of values previously reported using photographs, the the fact that 66% of respondents who could identify rip currents in photographs (Figure 3), but could not identify the actual rip current is of significant concern.

Third, our results suggest that the use of still images may not be an effective method to utilise in future research related to rip current identification. The selection of rip current images to use research surveys ultimately relies on subjectivity (Ménard

et al., 2018). Often the rip current is centred in these images, and thus clearly the focal point of the image, perhaps leading to increased identification bias. The survey design may also lead to participants trying to figure out the "correct" answer, perhaps based on text in previous questions or by obvious visual cues in the image, rather than providing an accurate overview of their perception of the hazard (Ménard et al., 2018). Researchers will often provide static images in which the rip current is obvious based on these visual cues, or taken from an elevated position to enhance the visualisation of the rip (e.g. Figure 3), instead

of from the beach or shoreline level, which is a more realistic viewpoint for someone making a decision on where to enter the water. The reality is that the surf zone is spatially and temporally dynamic and it often takes a prolonged period of observation to successfully identify rip currents. Identifying a rip current in a well-defined snapshot image versus in situ requires different skills and timeframes, and rarely do beachgoers seek an elevated position from which to observe rips prior to entering the water.





Finally, our findings also suggest that the use of still images may not be an effective method to use in public rip current education campaigns. One potential solution to this problem is to make use of the increasing availability of video footage of rip currents on social media, such as YouTube (Mackellar et al., 2015) and the rapidly increasing development and availability of 3D headsets and virtual reality, whereby people (e.g. school children) could interact with a 'live' and dynamic surf zone, including rip currents. These approaches are already being taken for water safety education in New Zealand (DPA, 2018), but

are yet to be used for quantitative analysis of surf zone hazard perception. Additionally, one approach gaining traction as a successful way of allowing people to visualise the rip current is the release of harmless dyes as a tracer (Brander et al., 2014) either in person or in video footage. The benefit here is that people have the opportunity to try and spot the rip current before, during, and after the dye release. As the dye fades, onlookers can try to look for the natural rip channel signature (i.e. darker gaps between breaking waves), and learn to associate this with the presence of a rip current.

## 4.3    Limitations

One of the limitations of this study is that it was conducted at a lifeguarded beach during guarded hours, and therefore there was a flagged bathing area, which was signposted as a safe place to swim. Therefore, respondents may automatically have associated the area outside of the flags to be dangerous, which potentially aided them in spotting the rip. Because this study was conducted in situ, familiarity with the beach may also have played a role in rip identification, with 31% of respondents

identifying Muriwai as their local beach. These individuals may have been reliant on previous knowledge of the beach to identify the likely rip location, which does not necessarily mean they would be able to identify a rip at an unfamiliar site. This may go some way towards explaining how some people who spotted no rip curents in photographs were able to identify the in situ rip. In the current study, 93% ($n = 27$) of the successful in situ identifications were made by people who lived in New Zealand, and 41% ($n = 12$) of the successful identifications were made by individuals who classed Muriwai as their local beach.

This study still represents an accurate depiction of a given beach population's ability to identify the rip, as beaches typically have a mix of visitors and locals with varying site-specific knowledge. Previous studies have shown that 40% of international visitors to New Zealand identified swimming at the beach as their most popular recreational activity whilst on holiday (Moran and Ferner, 2016).

    The surf zone is inherently dynamic due to factors such as tidal stage, individual wave sets, or changes in wind strength/direction

which all influence the degree to which the rip current was visible. As the surveys were conducted at different times across 7 different days, the appearance of the channel rip current may have changed, despite remaining in a persistent location during the study period. Therefore, it is possible that each respondent formulated their answer from slightly different viewpoints combined with different conditions. One potential solution to this problem is to make use of the increasing availability of 3D headsets and virtual reality, whereby each respondent could still interact with a 'live' and dynamic surf zone, but the experiment

could be better controlled such that each participant was given the exact same stimulus from which to formulate a response. Nevertheless, this study replicates the real-world conditions that people face when making decisions on the beach, and demonstrate that the selection of a safe place to swim without prolonged observation could equally result in the inadvertent selection of the rip current. This further highlights the difficulty in educational approaches where the main aim is to teach people to





identify the rip currents, given that conditions (and the visibility of the rip) change dramatically within a site during the course

of a day, let alone between sites.

## 5 Conclusions

Previous studies have reported rip identification rates based on respondents looking at photographs of rip currents, rather than rip currents in situ. This study represents the first attempt to examine relationships between an individual's beach experience, their ability to spot a rip in a photograph, and if this translated to an equal ability to locate a rip in situ at the beach. Overall,

78% of people were unable to identify a rip current at Muriwai, a high energy beach in New Zealand known for its pronounced channel rip currents. Respondents that were able to identify rip currents in photos were better able to identify the in situ rip current, but the majority of that group (66%) were still unable to translate this into meaningful identification of the in situ rip . Individuals that actively swim or surf at the beach were better able to identify rips when compared to those that never entered the water, or those that only waded in shallow water depths. Likewise, those that self-reported an increased water competence,

or those that had previously been caught in a rip current, were also more likely to be able to spot the in situ rip.

These results have major implications for the future use of photographs to assess beachgoers' ability to identify rip currents and for future rip current education strategies involving rip current identification. Many education programs use static imagery to 'teach' people to identify rip currents, but this study presents clear evidence that this skill does not translate usefully into in situ rip identification. Education either needs to make use of more immersive 3D/virtual reality strategies, videos of actual

rip current footage or dye releases to try and show people a dynamic surf zone, or continue to focus efforts on teaching people how to escape and cope with a rip current, given the knowledge that very few people are able to accurately spot one in situ.

*Author contributions.* All authors contributed to the conception of the study, KT conducted the fieldwork and initial analysis, SP conducted further analysis and compiled the first draft manuscript, and all authors contributed to the revision and refinement of the work through subsequent drafts.

*Competing interests.* The authors declare no competing interests.

*Acknowledgements.* KT was funded by a University of Canterbury Summer Scholarship, co-funded by Surf Life Saving New Zealand. Our thanks to the Muriwai lifeguards who hosted the research and provided consultation throughout the survey period.





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
