# Peer review of "Beachgoers' ability to identify rip currents at a beach in situ"

_Natural Hazards and Earth System Sciences, 2020_

## Referee Comment (RC1) · Anonymous Referee #1 · 20 Aug 2020

This study builds on previous work examining whether beachgoers are able to identify a rip current, and includes a new twist of asking beachgoers if they can spot a rip that is either in front of them or adjacent to them. The authors conclude that photographs are not necessarily a useful means for teaching beachgoers about rip currents and how to identify them in situ. While I believe that this is an important contribution to a growing body of literature on rip safety, I have some questions and concerns that need to be addressed before final publication:

* Is this phrase correct in the abstract: "only 34% were unable to translate this into a successful in situ rip identification,….". If I understand this correctly, "unable" should be "able".

* The photographs used in the survey are from above and at an angle to the beach.

[Figure]

This is very different from the photographs used in previous studies that were near perpendicular to the beach as if the beachgoer was standing along the back shore. How much of the lower accuracy in this study is associated with the orientation and perspective of the photograph versus ability to spot a rip? This builds on the perspective idea of Brannstrom et al. (2013) who noted that the NOAA rip current sign was designed with a perspective different from a beachgoer. This is noted in the discussion, but should be discussed further.

* Following from the above, the perspective of the photographs is different from those looking for the rip in situ, and each respondent would have had a different perspective of the rip based on their cross-shore and alongshore position. How would these differences affect the results? Can you provide some photographs of how the beach and surf would have appeared to the beachgoer taking the survey from the flagged area and from the area directly in front of the rip?

* Again following from the above, what was the spatial distribution of surveys on the beach relative to the in situ rip? Was there a difference in the ability of the beach user to spot the rip and/or identify a rip in the photograph based on their position on the beach? Essentially, were those sitting at or close to the rip able to spot the rip compared to those at a distance?

* The survey was only administered during times when wave breaking made the rip current visible by the breaking wave pattern. It would have been interesting to continue the questioning through the period when the rip was not active, albeit with a modified question, to determine if the "wrong" answers were consistent. This raises additional questions: * Were the breaking wave patterns and intensity consistent throughout the question period? If not, was there a difference in the ability of beachgoers to identify the rip based on wave and tidal conditions, and also based on their relative position? * Was the accuracy of the in situ questions worse at the start and end of the active rip period compared to at the peak? How did this vary by the distance of the respondent to the rip?

[Figure]

\* What was the distribution of answers on the photographs and how is an "X" on the photograph identified as correct or incorrect- I would assume the center of the X, but that should be described in more detail. Also- how large were the photographs shown to the beachgoer?

\* When a respondent was asked to identify the rip in situ, how were their answers determined to be correct or incorrect? For example, could they have been pointing in the right direction but for the wrong reason? Some anecdotal examples would be helpful in assessing the accuracy and validity of this question.

\* What was the sampling strategy for beachgoers and what was the rejection rate? How and where were beachgoers selected over the 7 days and over the period that the rip was active?

\* Were people told if their answers to the photograph question was correct or incorrect before being asked to identify the rip in situ? How would correcting their responses or not affect the ability of them to spot the rip? For example, if they were corrected about the location of the rip in the photograph, were they then using the photograph as an interpretive tool to find the rip in situ? If they were not corrected, it can't be argued that photographs are not useful since they were not used as an education device in the survey.

\* There is insufficient evidence to suggest a video or immersive experience is better for educating beachgoers since it was not directly tested. This statement should be qualified as needing further testing and not as a direct outcome of this study. Essentially, I don't think there is enough evidence to "advocate" at this time.

\* In addition to videos and immersive experiences, the authors should also consider whether there is a limit to education on spotting rips and whether other management strategies are more appropriate and impactful.

\* I think that more could be made about the results of Figure 5, which points to the disconnect between knowledge and behaviour. I am particularly surprised by the inability of those who "would not" swim beyond the flags to identify a rip. This is interesting and suggests a self-selection of beachgoers with limited knowledge to swim in the patrolled area, or was it just coincidence? If there was a difference in the number of people and respondents within the flagged area versus outside the flagged area, a Chi-square test would be useful to determine if the larger number is an indication of over- or under-representation by question.

In short, I think this is an interesting and important contribution, but there are a number of questions and concerns that need to be addressed in the manuscript. By number the concerns may appear major, but they should be easily discussed or qualified in the manuscript.

---

## Referee Comment (RC2) · Wayne Stephenson (Referee) · 7 Sep 2020

Overall this is a very useful study of beach hazard understanding by users. It highlights, rather worryingly, how poor people are at identifying rips on beaches, 78% is truly a worry. Also, the disjunct between ability to see rips in photographs used for education and what people see when they go to the beach is cause for concern. There are clearly important implications for beach safety, education and hazard mitigation that come from this work.

I have relatively little to suggest in the way of improvements, other than these minor points: Line 9 (abstract) unable – should be able. Title of section 2.2 – I found "Beach Survey design" a little confusing, since you are not surveying the beach (in the cross

shore profile sense), but beach user survey might be more accurate. I think a little more comment on the timing of the beach user survey would be useful, and which days of the week, was this Monday to Sunday – or Tuesday to Monday, Wednesday to Tuesday? Also what week in January was the survey undertaken? Given New Year and summer holidays in New Zealand, the cohort of people visiting the beach might different in the first week of January compared to those in the last week. I can imagine less experienced beach users being at the beach in early January, compared to late January. A comment on the representation of your surveyed users and how this might be different if you surveyed users in late February for example is worth considering. The frequency of beach visit data might look quite different in the later case. Given that the survey was run over 7 days – what were the wave and rip conditions? Did they change significantly over this period? Might it have been easier on some days to see a rip compared to others? You say the rip was prominent – but that is to an expert eye. Did changing wave conditions make the rip more or less prominent for participants? Given the sampling method, approaching people on the beach – this is not a true random sample, so I think some caution with regard to the statistical testing worth noting. Can you also report the number of refusals to participate? This helps to understand the randomness (or lack of) and size of sample. Chi square and T-tests are used are used with out reporting how it was decided to use parametric or non-parametric tests. I'd like to see better explanation and justification for the choice of tests. Replace photo with photograph throughout the manuscript. Replace didn't with did not. "spot" is often used with regard to participants ability to identify rip currents. I consider that rather informal, I suggest spot be changed to identify. Line 227 "This compares with.." How does it compare? Compares well? Compares poorly?

---

## Author Response (AR1)

**Authors' response to reviewers:** *NHESS-2020-244 Beachgoers' ability to identify rip currents at a beach in situ*

Dear Editor and Referees,

Thank you for your constructive comments on our manuscript entitled 'Beachgoers' ability to identify rip currents at a beach in situ'. We would like to thank you for your time, and feel that now having incorporated your comments that the manuscript is greatly improved as a result. In the response below we outline how we have engaged constructively with the referees' comments, and explicitly outline the revisions made to the manuscript.

Kind Regards,

Seb Pitman, on behalf of the co-authors.
* * *
**Editor Comments (EC):**

EC1: AR3 & AR4: You state that in response to these concerns, you have expanded Section 4.2, which discusses the implications for beach safety. Given the possible implication of the highlighted issue on your results, it seems useful to also acknowledge this issue in the results or limitation sections to ensure readers are aware of it.

We have expanded the second paragraph of the limitations to incorporate this explicitly. The text in bold italic below is new addition, whereas the normal text was existing but contributes towards this point also:

"Therefore, each respondent formulated their answer from slightly different viewpoints combined with different conditions. ***This study did not account for how viewing orientation, distance from the rip, or instantaneous hydrodynamic conditions impacted the ability to identify the rip.*** Any future in situ study should plan to incorporate some form of wave/tide measurement and a coastal imaging camera in order to make comparisons between identification rates and wave/tide heights and breaking wave patterns, ***as well as record observation locations relative to the rip***."

EC2: WS4: While I agree with WS that the t-test was not appropriate for your comparisons, I disagree with your choice to replace it with the Pearson chi-squared test. These are very different test that measure different things. While the former is a parametric test to examine differences in the means of an interval variable between two groups, the latter is used to examine differences between the expected and observed frequencies in a contingency table from categorical variables. However, your variables swimming competency, swimming distance, and swimming in the sea are all ordinal variables. Hence, a Wilcoxon rank-sum test would be able to indicate whether participants able to accurately identify the rip current were better or worse swimmers. This seems more in line with your original storyline and more insightful that just identifying differences in proportions without taking the ordinal character of the variable into account.

We have replaced these statistics with the Mann-Whitney U /Wilcoxon Rank Sum test, and amended the text as follows:

*"A Mann-Whitney U-test indicated maximum pool swim distance was significantly higher (median = 5, "200+ metres") among those that could identify a rip current compared to those that could not (median = 3, "51 - 100 m") (p = 0.006). Across all classes, self-reported*

*confidence about swimming that same distance at sea was significantly greater among males (median = 4, "Confident") than females (median = 3, "Unsure") (p = 0.004)."*

EC3: Presentation of statistical results (Section 3.1): I find your way of presenting proportions in the results section slightly confusion. For example, on line 157, you state "… higher proportions of males (33%, n = 15) that females (16%, n = 14) …". It was initially unclear to me whether the 33% was 5 of 15 or 15 of 46. In some sentences, it is more clear what your reference class is (e.g., Line 189: "Of the 29 respondents able to spot the rip current, …"), but I recommend that you change your format to something like "(33%, 15 of 46)", which avoid any misunderstandings.

We have updated this throughout the document.

EC4: Section 3.2: I find this paragraph quite confusing to read as you seem to be jumping around between different questions that you either interpret as interval or categorical variables. For example, you provide the descriptive statistics for the competence question on line 169, but you only mention that the difference is not statistically significant (which is fairly obvious) on line 174. I am also unclear on the reference class in your analysis of the confidence question. Did you include the responses from all participants or only the ones reporting that they were able to swim more than 100 m? If you used the entire sample, my calculations indicate that 32% among females (28 of 86) is significantly different from 54% among males (25 of 46) (chi-squared test: p = 0.025). I believe this paragraph would benefit from some clarifications.

We have re-arranged this paragraph to explicitly take each question in turn, outlining gender differences and the impact on rip spotting ability for each question before moving to the next. We have removed reference to a 'positive' response and instead are explicit in saying 70% of respondents replied 'Good' or 'Very Good'. We have also clarified that all respondents were asked about swimming that same distance at sea. The revised paragraph reads as follows:

*"Respondents were asked to self-rate their swimming competency through a series of questions (Table 4), with each response assigned a numerical value between 1 and 5, representing increasing competence. When asked to rate on a qualitative spectrum ('Poor', 'Fair', 'Good', 'Very Good'), 70% of respondents (n = 90 of 129) replied either 'Good' or 'Very Good' and there was no significant difference between genders. No significant difference was evident when ability to identify an in situ rip was analysed against self-reported estimates of swimming competence (Figure 4a). In order to qualify self reported competence, respondents were subsequently asked to estimate their maximum pool swimming distance. In response, 47% of females reported being able to swim in excess of 100 m, compared to 61% of males. A Mann-Whitney U-test indicated maximum pool swim distance was significantly higher (median = 5, "200+ metres") among those that could identify a rip current compared to those that could not (median = 3, "51 - 100 m") (p = 0.006). Across all classes, self-reported confidence about swimming that same distance at sea was significantly greater among males (median = 4, "Confident") than females (median = 3, "Unsure") (p = 0.004)."*

EC5: Fig. 4 to 6: It is unclear to me why the format and styling of your stacked bar charts changes between Fig 4, 5 and 6. Fig. 4 shows % respondents by gender on the y-axis, in Fig. 5 it is number of respondents, and in Fig. 6 it is normalized percentages of respondents. In addition, the styling of Fig. 6 is different from the other two. However, as far as I can tell, there is no difference in how the three different graphs are interpreted. Hence, I recommend a consistent format and presentation of these charts.

We have made Figs 4 – 6 consistent in style, including the y axis scale (percentage), and the colour of bars.
* * *
**Anonymous Referee (AR) 1:**

This study builds on previous work examining whether beachgoers are able to identify a rip current, and includes a new twist of asking beachgoers if they can spot a rip that is either in front of them or adjacent to them. The authors conclude that photographs are not necessarily a useful means for teaching beachgoers about rip currents and how to identify them in situ. While I believe that this is an important contribution to a growing body of literature on rip safety, I have some questions and concerns that need to be addressed before final publication:

AR1: Is this phrase correct in the abstract: "only 34% were unable to translate this into a successful in situ rip identification,. . ..". If I understand this correctly, "unable" should be "able".

Thank you - we corrected this by changing "unable" to "able".

AR2: The photographs used in the survey are from above and at an angle to the beach. This is very different from the photographs used in previous studies that were near perpendicular to the beach as if the beachgoer was standing along the back shore. How much of the lower accuracy in this study is associated with the orientation and perspective of the photograph versus ability to spot a rip? This builds on the perspective idea of Brannstrom et al. (2013) who noted that the NOAA rip current sign was designed with a perspective different from a beachgoer. This is noted in the discussion, but should be discussed further.

We corrected this by adding photograph results in Section 3.4 (Ln 218):

*"In this study, 31% (n = 41) of respondents were able to identify a rip in both photographs in Figure 3, 26% (n = 34) could identify a rip in only one photograph, and 43% (n = 57) were unable to identify a rip in either photograph."*

Further, we have also expanded the discussion in Section 4.2 (Ln 299) as follows:

*"This reinforces the findings of a study by Brannstrom et al. (2015) that showed a warning sign with a graphical representation of a rip current portrayed from an aerial view was useful in teaching people what to do if caught in a rip (swim parallel), but was not a useful means of helping people to identify a rip in situ. This goes someway to explain the disconnect outlined in this study between purely photograph based identifications (57% able to identify a rip in at least one image) and the translation of that ability into meaningful in situ identification at the beach, where only 22% of respondents could identify the rip.*

*[…..]*

*Ultimately, if photographs are to be used in further studies of rip identification, they should be site-specific and taken from a realistic beach perspective to ensure that beachgoers can situate themselves in place, rather than being asked to interpret a photograph taken from a viewpoint that bears no resemblance to the viewpoint afforded to them on the beach (Brannstrom et al., 2015; Ménard et al., 2018)."*

AR3: Following from the above, the perspective of the photographs is different from those looking for the rip in situ, and each respondent would have had a different perspective of the rip based on their cross-shore and alongshore position. How would these differences affect

the results? Can you provide some photographs of how the beach and surf would have appeared to the beachgoer taking the survey from the flagged area and from the area directly in front of the rip?

AR4: Again following from the above, what was the spatial distribution of surveys on the beach relative to the in situ rip? Was there a difference in the ability of the beach user to spot the rip and/or identify a rip in the photograph based on their position on the beach? Essentially, were those sitting at or close to the rip able to spot the rip compared to those at a distance?

This was beyond the scope our study so our investigation did not include collecting the information requested from these two related points. However, we acknowledge this is an important component of subsequent related studies involving rip identification. We have therefore included in Section 4.2 (Ln 310) the following recommendation:

*"The next logical step is to understand how spatial distribution of people on the beach influences their ability to identify the in situ rip current. Future studies should aim to identify how factors such as distance, and orientation of viewpoint relative to the main channel direction impact upon beachgoers ability to identify the rip."*

AR5: The survey was only administered during times when wave breaking made the rip current visible by the breaking wave pattern. It would have been interesting to continue the questioning through the period when the rip was not active, albeit with a modified question, to determine if the "wrong" answers were consistent. This raises additional questions:

AR6: Were the breaking wave patterns and intensity consistent throughout the question period? If not, was there a difference in the ability of beachgoers to identify the rip based on wave and tidal conditions, and also based on their relative position?

This comment was common from both reviewers (see comment WS3). The breaking wave heights over the study period (estimated from latest surf forecasts at the time) varied between 1.5 and 3 m (the average significant wave height at the site is 2.1m). Our analysis of lifeguard rescue data shows a disproportionate number of rip related rescues occurred when breaking wave heights were between 1.5 and 2.5 m. Therefore, the conditions during the study were representative of those which are of greatest concern to lifeguards in respect to rip current rescues. We unfortunately do not have the data to explicitly make comparisons based on wave/tide conditions related to individual responses. We have added this to Section 2.1 (Ln 104):

*"The breaking wave heights over the study period (estimated from latest surf forecasts at the time) varied between 1.5 and 3 m. Our own analysis of lifeguard rescue data shows a disproportionate number of rip related rescues occurred when breaking wave heights were between 1.5 and 2.5 m. Therefore, the conditions during the study were representative of those which are of greatest concern to lifeguards in respect to rip current rescues."*

AR7: Was the accuracy of the in situ questions worse at the start and end of the active rip period compared to at the peak? How did this vary by the distance of the respondent to the rip?

We unfortunately do not have the data to make these comparisons. In the limitations section 4.3 we discuss the dynamic nature of the surfzone, and how wave height and breaking conditions and tidal stage might influence the perspective. We have updated this section to incorporate the following recommendation for future work (Ln 347):

*"Any future in situ study should plan to incorporate some form of wave/tide measurement and a coastal imaging camera in order to make comparisons between identification rates and wave/tide heights and breaking wave patterns."*

AR8: What was the distribution of answers on the photographs and how is an "X" on the photograph identified as correct or incorrect- I would assume the center of the X, but that should be described in more detail. Also- how large were the photographs shown to the beachgoer?

We corrected this by clarifying in Section 2.2 (Ln 143) that the center of the 'x' was taken, as follows:

*"Participants were asked to draw an 'x' on the photograph to denote the location of the rip current. In assessing whether the answer was correct, the investigator would check that the centre of the x-mark corresponded to the darker area of the rip channel."*

The photographs were half an A4 portrait page wide, and the exact layout of the survey (and photograph sizes) can be seen in the supplementary material to the article. We unfortunately did not record the distribution of answers across the photographs, and are unable to retrieve this now as the University ethics approval required us to destroy returns after data had been summarised.

AR9: When a respondent was asked to identify the rip in situ, how were their answers determined to be correct or incorrect? For example, could they have been pointing in the right direction but for the wrong reason? Some anecdotal examples would be helpful in assessing the accuracy and validity of this question.

Verification was sought verbally in addition to a gesture. We have corrected this in the text in Section 2.2 (Ln 150) as follows:

*"In addition to pointing, in order to verify their answer and ensure accurate recording, participants were asked to describe the area in which they believed the rip to be located. Some participants responded by describing visual surfzone clues (e.g. the gap in the breaking waves) and some with landmarks (e.g. in front of the lifeguard tower) or distances (e.g. approximately 100 metres down the beach)."*

AR10: What was the sampling strategy for beachgoers and what was the rejection rate? How and where were beachgoers selected over the 7 days and over the period that the rip was active?

This comment was common from both reviewers (see comment WS4). We employed convenience sampling as a result of the relatively confined area within which we were operating and the requirement to maintain proximity to the rip channel in order to ensure it was visible to respondents. We have updated the methods section to reflect the convenience nature of sampling used. Refusal rate was not recorded, but our investigator qualitatively noted that more young males refused to participate, which is reflected in the gender bias. We have updated the text in Section 3 (Ln 160) as follows:

*"Although refusals were not recorded, our investigator qualitatively noted that a higher proportion of young males refused to participate."*

We subsequently expand upon the implications of this in our limitations in Section 4.3 (Ln 337) as follows:

*"Perhaps more significant in this study was the higher proportion of refusals to participate from young males. This is particularly pertinent as this demographic has been identified as at*

*risk in the global drowning literature (Woodward et al., 2013), and identified as a group more likely to over-estimate ability and under-estimate risk (Moran, 2011). Therefore, more work needs to be done to understand whether the previously reported under-estimation of risk is at all linked to an (in)ability to identify rip currents."*

AR11: Were people told if their answers to the photograph question was correct or incorrect before being asked to identify the rip in situ? How would correcting their responses or not affect the ability of them to spot the rip? For example, if they were corrected about the location of the rip in the photograph, were they then using the photograph as an interpretive tool to find the rip in situ? If they were not corrected, it can't be argued that photographs are not useful since they were not used as an education device in the survey.

We did not let people know if their photograph indications were correct or not prior to them attempting an in situ identification as we wanted to be sure that their knowledge base was the same as they attempted both tasks. We have clarified this in Section 2.2 (Ln 146) as follows:

*"Participants were not told whether or not their on paper rip identification was correct or not prior to attempting in situ identification such that their knowledge base was the same for both sets of identifications."*

With reference to education we were hoping to highlight that rip identification on paper does not equal an ability to do so in the real life, and therefore there remains a question over the utility of photos in education. We have replaced the following sentence in Section 4.2 (Ln 313):

*"Finally, our findings also suggest that the use of still images may not be an effective method to use in public rip current education campaigns."*

with

*"Finally, our findings suggest that more work is required to investigate whether photographs are actually a useful medium for rip current education campaigns, given the disconnect between successful identifications in photographs and real life."*

We have retained the following sentence in the conclusion (Ln 369):

*"These results have major implications for the future use of photographs to assess beachgoers' ability to identify rip currents and for future rip current education strategies involving rip current identification."*

as this does not imply that we tested this, purely that this study has implications for the use of photographs in education.

AR12: There is insufficient evidence to suggest a video or immersive experience is better for educating beachgoers since it was not directly tested. This statement should be qualified as needing further testing and not as a direct outcome of this study. Essentially, I don't think there is enough evidence to "advocate" at this time.

Thank you, instead we have highlighted other studies that focussed on education and advocated for videos to be used in place of images, in Section 4.3 (Ln 352) as follows:

*"Indeed, other studies have advocated for the use of video (Hatfield et al., 2012; Wilks et al., 2017) as a more appropriate means of visualising a rip current, and this would perhaps also allow for a more controlled measure of rip identification ability."*

We have amended the sentence in our conclusion (Ln 372) to highlight this as an area for research:

*"Future work should consider whether employing immersive 3D/virtual reality technologies and videos of actual rip current footage to present people with a dynamic surf zone would be a better means of educating people to identify rip currents."*

AR13: In addition to videos and immersive experiences, the authors should also consider whether there is a limit to education on spotting rips and whether other management strategies are more appropriate and impactful.

We acknowledge the current debate concerning the focus of rip current drowning prevention approaches, namely either preventative (teaching identification), or reactive (teaching escape strategies for those caught in a rip). We do not feel that we have yet exhausted education approaches, especially given the lack of investigation concerning video images. Much of the work on response to being caught highlights that people either tend to forget appropriate actions (Drozdzewski et al., 2012; 2015) or that there is significant variability in the success of various escape methods (McCarroll et al. 2014; van Leeuwen et al., 2016). This current lack of universally applicable escape strategies and the pre-disposition of those caught in a rip to panic and forget information highlights the importance of continued work in the education space, whilst other management strategies are developed and evaluated. As such we feel confident in the recommendation that this is presently an area requiring further work, albeit alongside other strategies which we already acknowledge in the introduction.

AR14: I think that more could be made about the results of Figure 5, which points to the disconnect between knowledge and behaviour. I am particularly surprised by the inability of those who "would not" swim beyond the flags to identify a rip. This is interesting and suggests a self-selection of beachgoers with limited knowledge to swim in the patrolled area, or was it just coincidence? If there was a difference in the number of people and respondents within the flagged area versus outside the flagged area, a Chi-square test would be useful to determine if the larger number is an indication of over- or underrepresentation by question.

In response to this suggestion we ran a chi square test grouping all those who would swim outside the flags and comparing it against those who would not, but it did not come back as significant ($X2 = 2.36$, $p = .125$). We have updated the text in Section 3.3 (Ln 211) to reflect that this test was done:

*"here was no statistically significant difference in rip spotting ability between those who chose to swim outside the flags and those who would not, although it does appear that many of those with lower knowledge may choose to remain between the flags."*

AR15: In short, I think this is an interesting and important contribution, but there are a number of questions and concerns that need to be addressed in the manuscript. By number the concerns may appear major, but they should be easily discussed or qualified in the manuscript.

Thank you for the constructive review and comments – we are pleased the manuscript has been recognised for the importance of the overall contribution, and are grateful for the improved the clarity as a result of this review.

**Wayne Stephenson (WS) Comments [Referee 2]:**

Overall this is a very useful study of beach hazard understanding by users. It highlights, rather worryingly, how poor people are at identifying rips on beaches, 78% is truly a worry. Also, the disjunct between ability to see rips in photographs used for education and what people see when they go to the beach is cause for concern. There are clearly important implications for beach safety, education and hazard mitigation that come from this work. I have relatively little to suggest in the way of improvements, other than these minor points:

WS1: Line 9 (abstract) unable – should be able.

Thank you - we corrected this by changing "unable" to "able".

WS1: Title of section 2.2 – I found "Beach Survey design" a little confusing, since you are not surveying the beach (in the cross shore profile sense), but beach user survey might be more accurate.

We have corrected this section heading as suggested to "beach user survey".

WS2: I think a little more comment on the timing of the beach user survey would be useful, and which days of the week, was this Monday to Sunday – or Tuesday to Monday, Wednesday to Tuesday? Also what week in January was the survey undertaken? Given New Year and summer holidays in New Zealand, the cohort of people visiting the beach might different in the first week of January compared to those in the last week. I can imagine less experienced beach users being at the beach in early January, compared to late January. A comment on the representation of your surveyed users and how this might be different if you surveyed users in late February for example is worth considering. The frequency of beach visit data might look quite different in the later case.

The study was conducted Tues-Monday in mid-January and based on our analysis of beach user numbers this was a representative period. Beach numbers are lower in early January, which we surmise is indicative of more local use, rising to a peak in the first two weeks of February indicative of visitors from further afield. In early January and over holiday periods beach visitation is more evenly spread throughout each day of the week, whereas away from these periods there are significant weekend spikes. Over the study period (mid-January), weekend spikes are prominent and therefore it was important to capture that period in the survey. We feel that given the beach user numbers for this period are representative of the average over the wider season we can infer the user demographic is representative, and have clarified the timing of the survey in the text in Section 2.2 (Ln 111) as follows:

*"…conducted over a 7-day period between Tuesday 15th and Monday 21st January 2019 (Austral Summer). Beach visitor numbers in mid-January are representative of the wider season typically averaging around 100 people at any given time during the week and 800 during the weekend. Visitation numbers steadily increase towards the second week of February, peaking at around 2500 at any given point over the weekend."*

WS3: Given that the survey was run over 7 days – what were the wave and rip conditions? Did they change significantly over this period? Might it have been easier on some days to see a rip compared to others? You say the rip was prominent – but that is to an expert eye. Did changing wave conditions make the rip more or less prominent for participants?

This comment was common from both reviewers (see comment AR5 and AR6). The breaking wave heights over the study period (estimated from latest surf forecasts at the time) varied between 1.5 and 3 m (the average significant wave height at the site is 2.1m). Our analysis of lifeguard rescue data shows a disproportionate number of rip related rescues occurred when breaking wave heights were between 1.5 and 2.5 m. Therefore, the conditions during the study were representative of those which are of greatest concern to lifeguards in respect to rip current rescues. We unfortunately do not have the data to explicitly make comparisons based on wave/tide conditions related to individual responses. We have added this to Section 2.1 (Ln 104):

*"The breaking wave heights over the study period (estimated from latest surf forecasts at the time) varied between 1.5 and 3 m. Our own analysis of lifeguard rescue data shows a disproportionate number of rip related rescues occurred when breaking wave heights were between 1.5 and 2.5 m. Therefore, the conditions during the study were representative of those which are of greatest concern to lifeguards in respect to rip current rescues."*

We acknowledge that the variability in rip prominence is not something we have been able to quantify. In the limitations section 4.3 (Ln 342) of the study we address this explicitly as follows:

*"The surf zone is inherently dynamic due to factors such as tidal stage, individual wave sets, or changes in wind strength/direction which all influence the degree to which the rip current was visible. As the surveys were conducted at different times across 7 different days, the appearance of the channel rip current may have changed, despite remaining in a persistent location during the study period. Therefore, it is possible that each respondent formulated their answer from slightly different viewpoints combined with different conditions. One potential solution to this problem is to make use of the increasing availability of 3D headsets and virtual reality, whereby each respondent could still interact with a 'live' and dynamic surf zone, but the experiment could be better controlled such that each participant was given the exact same stimulus from which to formulate a response. Nevertheless, this study replicates the real-world conditions that people face when making decisions on the beach, and demonstrate that the selection of a safe place to swim without prolonged observation could equally result in the inadvertent selection of the rip current."*

WS4: Given the sampling method, approaching people on the beach – this is not a true random sample, so I think some caution with regard to the statistical testing worth noting. Can you also report the number of refusals to participate? This helps to understand the randomness (or lack of) and size of sample. Chi square and T-tests are used are used without reporting how it was decided to use parametric or nonparametric tests. I'd like to see better explanation and justification for the choice of tests.

This comment was common from both reviewers (see comment AR10). It is true that this is not a random sample representative of the entire population. We have reflected this in the title of the manuscript by referring to 'Beachgoers' ability to identify a rip current – acknowledging that this is a subset of the population. Within this subset, the main skew is towards females. Whilst refusals were not recorded, our investigator qualitatively noted that males were more likely to refuse to participate, and we have added this into our results (Section 3, Ln 159) as follows:

*"The sample population is gender-imbalanced, with 65% of respondents being female. Although refusals were not recorded, our investigator qualitatively noted that a higher proportion of young males refused to participate."*

We have also updated the limitations section 4.3 (Ln 337) to reflect this, as follows:

*"Perhaps more significant in this study was the higher proportion of refusals to participate from young males. This is particularly pertinent as this demographic has been identified as at risk in the global drowning literature (Woodward et al., 2013), and identified as a group more likely to over-estimate ability and under-estimate risk (Moran, 2011). Therefore, more work needs to be done to understand whether the previously reported under-estimation of risk is at all linked to an (in)ability to identify rip currents."*

With regard to statistical tests we have opted to replace the t-tests with non-parametric Mann-Whitney U tests, given that they do not require assumptions about the population characteristics in the same way that parametric tests do. We have updated the results to reflect this, with the only difference being that there is no longer a statistically significant difference in rip spotting ability based on self-reported qualitative swimming ability. There remains statistical significance between rip spotting ability and quantitative swimming competence in the form of maximum pool swimming distance.

WS5: Replace photo with photograph throughout the manuscript. Replace didn't with did not. "spot" is often used with regard to participants ability to identify rip currents. I consider that rather informal, I suggest spot be changed to identify.

We have implemented all of these corrections.

WS6: Line 227 "This compares with.." How does it compare? Compares well? Compares poorly?

We have corrected this to reflect that it is a "good comparison" to other studies.

[revised manuscript text omitted]